# CausalStructCodec: Causally-aware observational and interventional data generator

**Louis Hernandez**
Craft.AI
`louis.hernandez@craft.ai`

**Matthieu Boussard**
Craft.AI
`matthieu.boussard@craft.ai`

## Abstract

The main Goal of causal generative models is to generate observational, interventional and counterfactual data. They are based on typical data generation architectures, such as VAEs, GANs etc. However, a recent data generation architecture, named Codecs, is more computationally efficient and allows for complex data generation. In this work, we introduce the CausalStructCodecs (CSC), a novel causally-aware architecture based on a specific codec, the StructCodec. We show that results for non-complex data are level with state-of-the-art models for observational and interventional data generation, in significantly fewer epochs.

## 1 Introduction

Generative models are useful tools for data augmentation or private data sharing. In the realm of causality, Structural Causal Models (SCM)(Pearl, 2009) are theoretical models representing the true data-generating process. They allow for the computation of interventional and counterfactual generation. But these models are theoretical and, in practice, we do not have access to the functions of the SCM that truly generate the data. We can however have access to the causal graph (Pearl, 2009), using domain knowledge. Dedicated generative models have emerged to leverage that knowledge of the causal graph to generate data, mimicking an SCM, such as MultiCVAE (Karimi et al., 2020), CAREFL (Khemakhem et al., 2021) or VACA (Sanchez-Martin et al., 2021). However, we found that these models have limitations like computational efficiency and lack of flexibility regarding complex data structures. Therefore, we provide the CausalStructCodecs (CSC), a novel architecture based on Codecs(Canale et al., 2022), designed to tackle these issues. We show that it generates interventional and observational data with at least equivalent accuracy to state-of-the-art models while taking advantage of the performance and flexibility of the Codecs architecture.

## 2 The architecture

Like other Codecs, CSC can be defined as a quadruplet $C = (E, D, S, L)$, defining an encoder $E$, a decoder $D$, a Sampler $S$ and a loss $L$, see Figure 1. While details will be explained in the next paragraphs, for the sake of conciseness, we invite the reader to refer to Canale et al. (2022) for details about the Codecs. Our contributions are two-fold, as indicated in figure 1, firstly (**1**, in figure 1), we remove the transformer in the encoder, and replace it partly with a MLP for the embedding, secondly (**2**, in figure 1), we use a mask on the decoder of the transformer to take into account the causal graph. We provide an algorithm for interventional sampling as well.

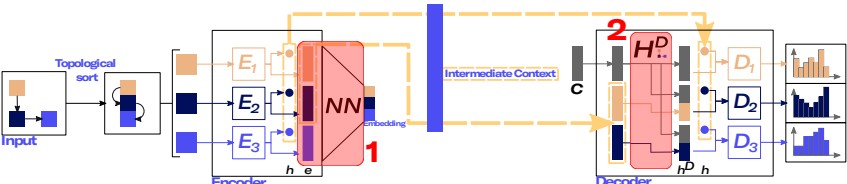

Figure 1: Architecture of the CausalStructCodec

**Pre-treatment** Let $(x_1, ..., x_n)$ be an input composed of categorical and numerical features, with its causal graph $\mathcal{G}$ (the same for every data point). Like a StructCodec (Canale et al., 2022), CSC processes features one after the other, and can not process a child without having processed their parent. Thus, the features are topologically sorted according to $\mathcal{G}$ before being used as input to the Encoder. However, unlike CAREFL, which uses only the topological sort and not the full DAG, the full information of the causal graph is retained and used in the transformers.

**Encoder** First, the CSC encoder $E$ calls the encoders of the sub codecs $E_k$ to process each feature $k$ individually and returns for each $k$, $E_k(x_k) = (e_k, h_k)$ with $e_k$ the encoding and $h_k$ the intermediate context. While the original StructCodecs uses a transformer in its encoder, we do not use it here. Indeed, it would be redundant with the other transformer, and induce computation in contradiction to the causal DAG. To solve this issue, we add a Dense Neural Network ($NN$ Figure 1) to output an embedding, which will not be used here but might be when CSC is used as a sub-structure for larger codecs, such as a ListCodec, a StructCodec, or even CSC itself. Then, unlike in the original StructCodecs, the intermediate context used by the decoder is $\{e_i\}_{i \notin T(\mathcal{G})}$, where $T(\mathcal{G})$ is the set of terminal nodes of $\mathcal{G}$, along with the individual intermediate context $h$.

**Decoder** The decoder uses the non-terminal nodes as input for a causal transformer, defined as self-attention layers with causal masking, $H^D$. This transformer takes advantage of the causal graph to ensure that for each $k$, the decoded distribution of $x_k$ is only computed from the parents and from the conditioning vector $c$, a vector used as an input for every codec, that only varies when the codec is used as a sub-structure. This step can be written as $h_k^D = H^D(c, \{e_i\}_{i \in pa(\mathcal{G}, k)})$, where $pa(\mathcal{G}, k)$ is the set of parents of $k$ in $\mathcal{G}$. Finally, this last vector $h_k^D$ is used as a conditioning vector for the final decoders, with the intermediate context $h_k$, to obtain the final distribution representation $d_k = D_k(h_k^D, h_k)$.

**Sampler** After being trained following the previous steps, CSC uses its sampler to generate interventional and observational data. The variables are sampled autoregressively. The first feature is sampled from $d_1 = H^D(c)$, then the last sampled feature $x_k$ is encoded as $(e_k, h_k)$, which goes through the whole model again to obtain its children in the causal graph. To simulate an intervention $do(x_k)$, instead of sampling $x_k$ from the intervention obtained from its parents, one can simply set it to a value, and let that value run through the model again.

**Loss** The loss can be computed using $d_k$ and $x_k$, using $L = -\log(P(x_k|d_k))$.

## 3 RESULTS

We evaluated the capacity of our model to generate interventional and observational data using synthetic and semisynthetic datasets from Sanchez-Martin et al. (2021). We found that in all tasks, our model performs similarly to VACA, and better than CAREFL and MultiCVAE, as can be seen in Table 1. Additionally, following Canale et al. (2022), we have reasons to believe our method is more computationally efficient, which is in part confirmed by the low number of epochs necessary to reach the best performance here.

| MMD | loan | | collider | | mgraph | | Number of epochs |
|---|---|---|---|---|---|---|---|
| | Obs | Int | Obs | Int | Obs | Int | |
| MultiCVAE | $5.97 \pm 0.21$ | $11.25 \pm 0.19$ | $1.04 \pm 0.21$ | $14.67 \pm 0.19$ | $1.32 \pm 0.12$ | $4.07 \pm 0.11$ | $448.3 \pm 0.31$ |
| CAREFL | $2.19 \pm 0.12$ | $3.42 \pm 0.11$ | $2.77 \pm 0.22$ | $1.99 \pm 0.20$ | $5.33 \pm 0.32$ | $4.96 \pm 0.29$ | $291.93 \pm 49.91$ |
| VACA | $\mathbf{0.64 \pm 0.05}$ | $\mathbf{1.07 \pm 0.04}$ | $0.41 \pm 0.06$ | $\mathbf{0.29 \pm 0.06}$ | $0.20 \pm 0.04$ | $\mathbf{0.69 \pm 0.04}$ | $373.9 \pm 22.56$ |
| CSC | $1.18 \pm 0.05$ | $2.34 \pm 0.04$ | $\mathbf{0.31 \pm 0.06}$ | $0.45 \pm 0.06$ | $\mathbf{0.17 \pm 0.03}$ | $0.70 \pm 0.03$ | $\mathbf{7.67 \pm 1.35}$ |

Table 1: Comparison of CSC with other causally-aware generators (datasets detailed appendix?B)

## 4 CONCLUSION

We introduced a novel architecture for causal data generation and performances that are equivalent to or better than state-of-the-art models. Future work could include causal discovery, like CAREFL in Khemakhem et al. (2021). Furthermore, alternative transformers could potentially improve the results presented here. Finally, this model is still missing a method to generate counterfactual data.

URM STATEMENT

The authors acknowledge that the first author of this work meets the URM criteria of the ICLR 2023 Tiny Papers Track. As a young (25 years old) master's student, that does not have and is not pursuing a PhD, we believe the first author of this work meets the URM criteria. Furthermore, this work has been done in a small research team (3 people, only one with a PhD) in a start-up, with the first author of this work being the only one focused on causality.

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

## A    EXPERIMENTAL PROTOCOL

The results in Table 1 are obtained following an experimental protocol following Sanchez-Martin et al. (2021). We compare our method with the existing state-of-the-art methods VACA, CAREFL and MultiCVAE. The datasets used here are the non-additive M-Graph, the linear collider, and the loan dataset, inspired by the German dataset (Dua & Graff, 2017). In practice, we use the official implementation of VACA to run the experiments on the concurrent models and run our own experiments on the same training, validation and testing datasets. CSC was implemented in JAX, and the implementation is available on `https://github.com/craft-ai/causal-struct-codec` after the double-blind review process. For each intervention, we follow the interventions implemented for the VACA paper. We run each experiment, both interventional and observational for ten different seeds. Therefore, the results in Table 1 are the average of these ten runs for observational generation, and for the interventional distribution, it is the average of the ten runs for each of the interventions, so a total of ten times the number of interventions for each dataset: 8 for collider, 14 for loan and 8 for M-graph.

## B    DATASETS

In this section, we detail the datasets used, taken from Sanchez-Martin et al. (2021).

### B.1    LOAN

The loan dataset is a synthetic dataset inspired by the German dataset Dua & Graff (2017). It consists of 7 variables : gender $G$, age $A$, education $E$, loan amount $L$, loan duration $D$, income $I$

and savings $S$.

$$G = U_G$$
$$A = -35 + U_A$$
$$E = -0.5 + \frac{1}{1 + e^{1-0.5G-(1+e^{-0.1A})^{-1}}} - U_E$$
$$L = 1 - 0.01(A - 5)^2 + G + U_L$$
$$D = -1 + 0.1A + 2G + L + U_D$$
$$I = -4 + 0.1(A + 35) + 2G + GE + U_I$$
$$S = -4 + 1.5\mathbb{I}_{\{I>0\}}I + U_S$$

With $U_G \sim$ Bernoulli$(0.5)$, $U_A \sim$ Gamma$(10, 3.5)$, $U_E \sim \mathcal{N}(0, 0.25)$, $U_L \sim \mathcal{N}(0, 4)$, $U_D \sim \mathcal{N}(0, 9)$, $U_S \sim \mathcal{N}(0, 25)$ and $U_I \sim \mathcal{N}(0, 4)$. Its causal graph is shown on Figure 2.

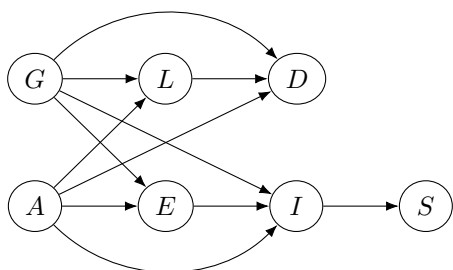

Figure 2: Causal graph for the loan dataset

## B.2  COLLIDER

We use a collider synthetic dataset, in its linear variant. Its equations are :

$$X_1 = U_1$$
$$X_2 = U_2$$
$$X_3 = 0.05X_1 + 0.25X_2 + U_3$$

With $U_E \sim 0.5\mathcal{N}(-2, 1.5) + 0.5\mathcal{N}(1.5, 1)$, $U_2 \sim \mathcal{N}(0, 1)$ and $U_2 \sim \mathcal{N}(0, 1)$. Its causal graph is shown on Figure 3.

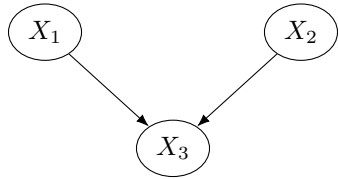

Figure 3: Causal graph for the collider dataset

## B.3  M-GRAPH

Lastly, we use the non-additive variant of the M-Graph dataset. The equations are :

$$X_1 = U_1$$
$$X_2 = U_2$$
$$X_3 = X_1 \times U_3$$
$$X_4 = (-X_2 + 0.5 \times (X_1)^2)U_4$$
$$X_5 = -1.5(X_2)^2 U_5$$

With, $\forall i \in 1, ..., 5, U_i \sim \mathcal{N}(0, 1)$. Its causal graph is shown on Figure 4.

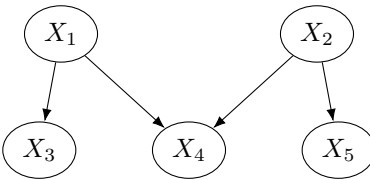

Figure 4: Causal graph for the M-Graph dataset

