# OpenReview forum: "CausalStructCodec: Causally-aware observational and interventional data generator"
_ICLR.cc/2023/TinyPapers — Submitted to Tiny Papers @ ICLR 2023_

### Official Review · Reviewer_UuNC · 2023-03-29

**Confidence:** 3

**Summary Of Contributions:**

This paper introduces an architecture for causal data generation based on the architecture Codecs, which is more computationally efficient.

**Rating:**

Needs Clarification (NC): a submission which does not meet the reviewing criteria and needs clarification for its described problem or solution

**Strengths And Weaknesses:**



Strengths:
1. Provide convincing empirical results to show the efficiency of the proposed approach.

Weakness:
1. The contribution and the novelty are not clear enough for this paper.
2. The proposed approach incorporates the previous framework with slight modifications on encoder and decoder without illustrations on motivations, more analysis should be provided.



**Suggested Changes:**

1. The writing of this paper should be improved, with more motivations and discussions for the proposed approach.
2. The proposed approach is relatively incremental for current version.

---

### Official Review · Reviewer_okbr · 2023-04-01

**Confidence:** 4

**Summary Of Contributions:**

This paper presents a novel architecture for a casual generative model-- an adaptation of  Canale et al.  with a directed acyclic graph (DAG) processing capability as an encoder input. The DAG captures casual relationships among feature variables.

**Rating:**

High Potential (HP): a submission which meets the reviewing criteria and has potential to make an impact on the field

**Strengths And Weaknesses:**

Strength
1. The author's claims of a computationally efficient method are supported by empirical results. The findings show that the model requires fewer epochs to achieve the best result.
2. Findings are communicated clearly.

Weakness
1. The final distribution from the decoder is obtained by combining the intermediate representations of the encoder and the hidden states of the decoder. The authors did not mention the intuition of their design modification.
2. Reproducibility: The reader can reproduce the design. However, the authors did not share the code.






**Suggested Changes:**

Please provide the source code link.

---

### Author Response · Authors · 2023-05-31
**Archival opt-in**

We would like to thank all reviewers for their evaluation of our work. After taking into account their feedbacks, we believe our paper is ready to be archived, and we would like to opt-in for its archival.

---

### Comment · Area_Chair_mPNL · 2023-06-06

This work meets the threshold for archival, contents the URM statement and is deanonymized

---

### Meta-Review · Area_Chair_mPNL · 2023-04-04

**Recommendation:** Invite to archive
**Confidence:** 2

**Metareview:**

The authors present an algorithm (an adaption of Canale et al. 2022) for data generation in a causally aware manner. The reviewers are split: one (confidence 3/5) rates this paper quite low as "Needs clarification" mostly citing a lack of novelty. The other (confidence 4/5) rates this paper as having "High potential" citing good empirical support for the authors claim.

The work is significant as it upgrades an existing generative model ("Codecs") by exploiting knowledge of the directed acyclic graph (DAG) present in generative process (specifying causality between features) of true data to make a "causally-aware" Codecs-based generative model which train faster than other causally-aware non-Codecs-based model.

The work is explained somewhat clearly but it could be improved to lower the barrier to understanding for non-domain-expert readers.

Pros:
* Empirically supported claim that their model trains to an equal level, but faster than, other comparable models.

Cons:
* Lack of clear communication about the authors specific contribution.
* High barrier to entry from understanding this contribution; albeit the authors do point to the relevant literature and the paper is constrained to two pages.
* No comparison of their results to the original Codec or StructCodec models. Perhaps this is because it would likely perform worse due to replacement of the transformer with a DNN. (This is not a criticism, just a speculation worth clarifying).

**Summary:**

The Codecs generative model is updated to account for causal structure in the input features. This "causally-aware" codecs model trains faster than other comparable causally-aware models. Reviewers thought the empirical results were convincing but wanted better explanations of the motivation and clearer descriptions of the novelty.

**Comments And Feedback To The Authors:**

One suggestion I would make is to adapt some phrasing for non-domain expert readers. What does it mean to generate data in a causally-aware manner? Is there a simple example of something a causally aware data generator would solve but a non-causally are one wouldn't. Such an example would go a long way to clarifying the contribution of this paper.

Table 1 is poorly explained: what is loan, collider, graph etc.?

Figure 1 caption should say exactly how this model differs from StructCodecs. It would be particularly nice if the parts where it differs were coloured differently, to make this clearer.

Typo:
* Additionnally

**Reason For Not Giving A Higher Recommendation:**

The authors could communicate their findings more clearly in a way which requires less domain-specific knowledge.

**Reason For Not Giving A Lower Recommendation:**

In my opinion reviewer UuNC does not give sufficient justification for their low score. There are clearly some interesting and novel contributions here.

---

> ### Author Response · Authors · 2023-05-31
> **Response to meta-review by Area Chair mPNL**
>
> Thank you for your very complete feedback. We have revised our work to take your feedback into account as good as we could while staying within the two pages limit.
>
> * We modified figure 1 and the text around it to better explain our contribution
> * We added appendix to clearly explain the data in table 1
> * Some other minor reformulations to try to make it clearer to non-domain expert, although the lack of space made it particularly hard.

---

### Decision · Program_Chairs · 2023-04-10

Invite to archive